# Causal neural mechanisms of context-based object recognition

**Miles Wischnewski[1,2], Marius V Peelen[1]\***

[1]Donders Institute for Brain, Cognition and Behaviour, Radboud University, Nijmegen, Netherlands; [2]Department of Biomedical Engineering, University of Minnesota, Minneapolis, United States

**Abstract** Objects can be recognized based on their intrinsic features, including shape, color, and texture. In daily life, however, such features are often not clearly visible, for example when objects appear in the periphery, in clutter, or at a distance. Interestingly, object recognition can still be highly accurate under these conditions when objects are seen within their typical scene context. What are the neural mechanisms of context-based object recognition? According to parallel processing accounts, context-based object recognition is supported by the parallel processing of object and scene information in separate pathways. Output of these pathways is then combined in downstream regions, leading to contextual benefits in object recognition. Alternatively, according to feedback accounts, context-based object recognition is supported by (direct or indirect) feedback from scene-selective to object-selective regions. Here, in three pre-registered transcranial magnetic stimulation (TMS) experiments, we tested a key prediction of the feedback hypothesis: that scene-selective cortex causally and selectively supports context-based object recognition before object-selective cortex does. Early visual cortex (EVC), object-selective lateral occipital cortex (LOC), and scene-selective occipital place area (OPA) were stimulated at three time points relative to stimulus onset while participants categorized degraded objects in scenes and intact objects in isolation, in different trials. Results confirmed our predictions: relative to isolated object recognition, context-based object recognition was selectively and causally supported by OPA at 160–200 ms after onset, followed by LOC at 260–300 ms after onset. These results indicate that context-based expectations facilitate object recognition by disambiguating object representations in the visual cortex.

**\*For correspondence:**
m.peelen@donders.ru.nl

## Introduction

Objects are typically seen within a rich, structured, and familiar context, such as cars on a road and chairs in a living room. Decades of behavioral work have shown that context facilitates the recognition of objects (*Bar, 2004*; *Biederman et al., 1982*; *Oliva and Torralba, 2007*). This contextual facilitation is crucial for everyday behavior, allowing us to recognize objects under poor viewing conditions (*Figure 1*), at a distance, in clutter, and in the periphery where visual resolution is low. Yet despite the pervasive influence of context on object recognition, our knowledge of the neural mechanisms of object recognition almost exclusively comes from studies in which participants view clearly visible isolated objects without context. These studies have shown that isolated object recognition results from the transformation of local, low-level features into view-invariant object representations along the ventral stream (*DiCarlo et al., 2012*; *Liu et al., 2009*; *Riesenhuber and Poggio, 1999*; *Serre et al., 2007*). Does a similar local-to-global hierarchy support context-based object recognition?

One possibility is that context-based object recognition is supported by the parallel feedforward processing of local object information in the ventral stream object pathway and global scene processing in a separate scene pathway (*Henderson and Hollingworth, 1999*; *Park et al., 2011*). Output

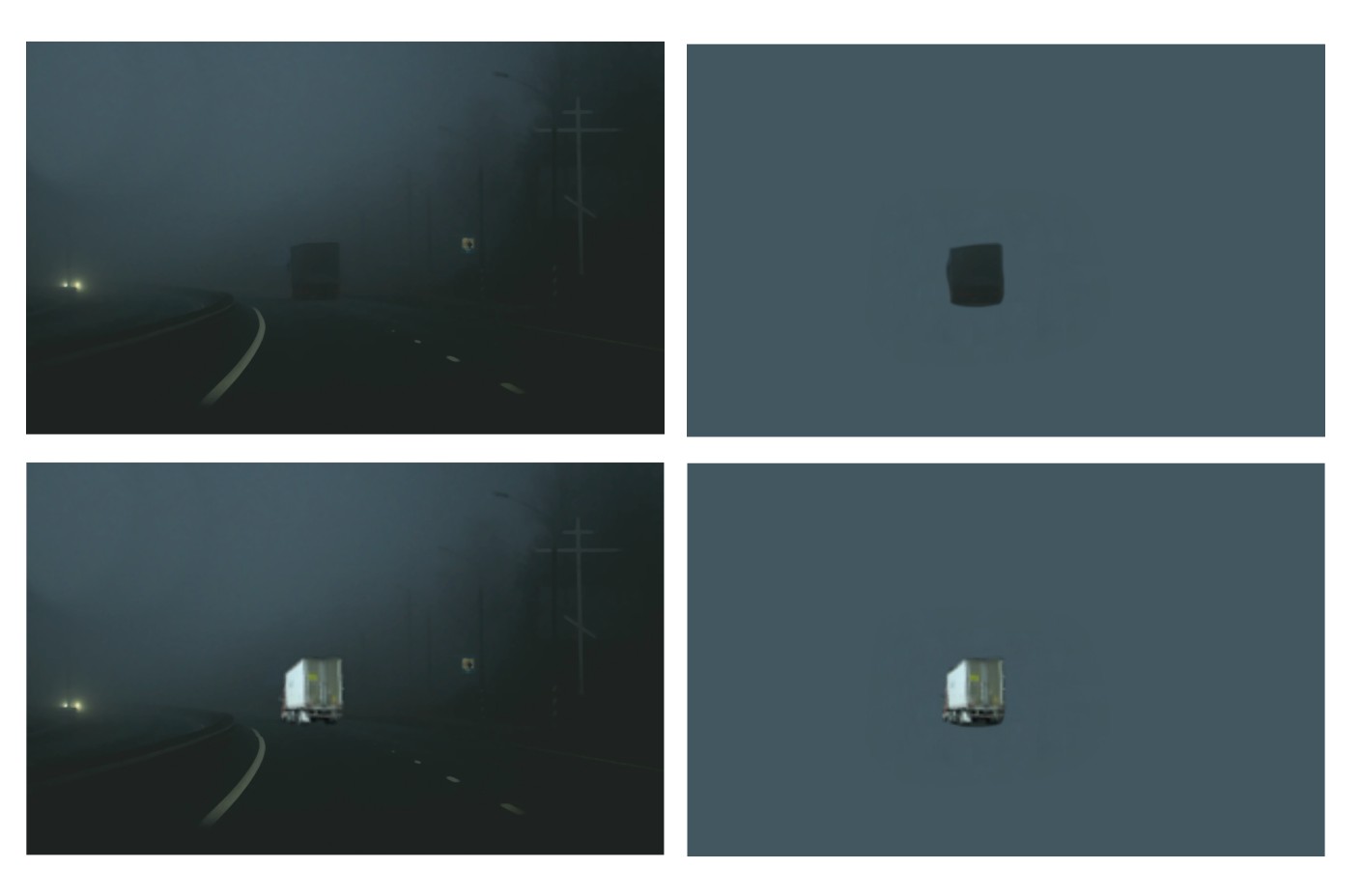

**Figure 1.** Example of context-based object recognition. At night (top panels), the truck is easily recognized by participants when placed in context (left) but not when taken out of context (right). With sufficient light (bottom panels), the truck is easily recognized also when presented in isolation.

of these pathways may then be combined in downstream decision-making regions, leading to contextual benefits in object recognition. Alternatively, context-based object recognition may be supported by feedback processing, with scene context providing a prior that is integrated with ambiguous object representations in the visual cortex (*Bar, 2004*; *Brandman and Peelen, 2017*; *de Lange et al., 2018*). Neuroimaging studies have not been able to distinguish between these possibilities because the contextual modulation of neural activity in object-selective cortex (*Brandman and Peelen, 2017*; *Faivre et al., 2019*; *Gronau et al., 2008*; *Rémy et al., 2014*) could precede but also follow object recognition, for example reflecting post-recognition imagery (*Dijkstra et al., 2018*; *Reddy et al., 2010*).

To distinguish between these accounts, we used transcranial magnetic stimulation (TMS) to interfere with processing in right object-selective lateral occipital cortex (LOC; *Grill-Spector, 2003*; *Malach et al., 1995*) and right scene-selective occipital place area (OPA; *Dilks et al., 2013*; *Grill-Spector, 2003*) at three time points relative to stimulus onset. We additionally stimulated the early visual cortex (EVC) to investigate the causal contribution of feedback processing in this region during both isolated object recognition and context-based object recognition (*Camprodon et al., 2010*; *Koivisto et al., 2011*; *Pascual-Leone and Walsh, 2001*; *Wokke et al., 2013*). EVC stimulation was targeted around 2 cm above the inion, with the coil positioned such that TMS induced static phosphenes centrally in the visual field, where the stimuli were presented. This region corresponds primarily to V1 (*Koivisto et al., 2010*; *Pascual-Leone and Walsh, 2001*). The three regions were stimulated in separate pre-registered experiments (N=24 in each experiment; see Materials and methods).

Because TMS effects are variable across individuals, for example, due to individual differences in functional coordinates but also skull thickness and subject-specific gyral folding patterns (*Opitz et al., 2013*), we used a TMS-based assignment procedure to ensure the effectiveness of TMS over each of the three stimulated regions at the individual participant level (*van Koningsbruggen et al., 2013*). To achieve this, all 72 participants in the current study first underwent a separate TMS session in which the effectiveness of TMS over the three regions was established using object and scene recognition tasks (for the full procedure and results of this screening experiment, see *Wischnewski and Peelen, 2021*). Only participants who showed reduced scene recognition performance after OPA stimulation were assigned to the OPA experiment (N=24), only participants who showed reduced object recognition performance after LOC stimulation were assigned to the LOC experiment (N=24), and only participants who experienced TMS-induced phosphenes after EVC stimulation were assigned to the EVC experiment (N=24). All 72 participants satisfied at least one of these criteria such that no participants had to be excluded.

In all experiments, participants performed an unspeeded eight-alternative forced-choice object recognition task, indicating whether a briefly presented stimulus belonged to one of the eight categories (*Figure 2*). Participants performed this task for clearly visible isolated objects (*isolated object recognition*) as well as for degraded objects presented within a congruent scene context (*context-based object recognition*). In addition to the object recognition tasks, participants also performed a scene-alone task in which the object was cropped out and replaced with background. In this condition, participants had to guess the object category of the cropped-out object.

Predictions (*Figure 3a*) were based on the findings of recent functional magnetic resonance imaging (fMRI) and magnetoencephalography (MEG) experiments investigating context-based object recognition (*Brandman and Peelen, 2017*). In those experiments, participants viewed degraded objects in scene context, degraded objects alone, and scenes alone. Behavioral results showed that the degraded objects were easy to recognize when presented in scene context (>70% correct in a nine-category task) but hard to recognize when presented alone (37% correct). fMRI results showed that the multivariate representation of the category of the degraded objects in LOC was strongly enhanced when the objects were viewed in scene context relative to when they were viewed alone. Importantly, the corresponding scenes presented alone did not evoke discriminable object category responses in LOC, providing evidence for supra-additive contextual facilitation. Interestingly, the contextual facilitation of object processing in LOC was correlated with concurrently evoked activity in scene-selective regions, suggesting an interaction between scene- and object-selective regions. MEG results showed that the information about the category of the degraded objects in scenes (derived from multivariate sensor patterns) peaked at two time points: at 160–180 ms and at 280–300 ms after stimulus onset. Crucially, only the later peak showed a significant contextual facilitation effect, with more information about the degraded objects in scenes than the degraded objects alone. Similar to the LOC results, at this time point, the scenes alone did not evoke discriminable object category responses, such that the contextual facilitation of object processing could not reflect the additive processing of scenes and objects. Taken together, these results indicate that scenes—processed in scene-selective cortex—disambiguate object representations in LOC at around 300 ms after stimulus onset.

The current TMS study was designed to provide causal evidence for this account. Differently from the neuroimaging studies, here we compared the recognition of degraded objects in scenes with the recognition of intact objects alone, rather than degraded objects alone. This was because the large accuracy difference between the recognition of degraded objects in scenes and degraded objects alone prevents a direct comparison of TMS effects between these conditions. Furthermore, this design allowed us to compare the causal neural mechanisms underlying object recognition based on scene context and local features, with the possibility to match the tasks in terms of recognition accuracy.

## Results

Across all TMS conditions, objects were equally recognizable when presented in isolation (without degradation; 75.8%) and when presented degraded within a scene (76.7%; main effect of Task: $F_{(1,71)}=1.22$, $p=0.27$), showing that scene context can compensate for the loss of object visibility induced by the local degradation (*Brandman and Peelen, 2017*). Importantly, despite the equal

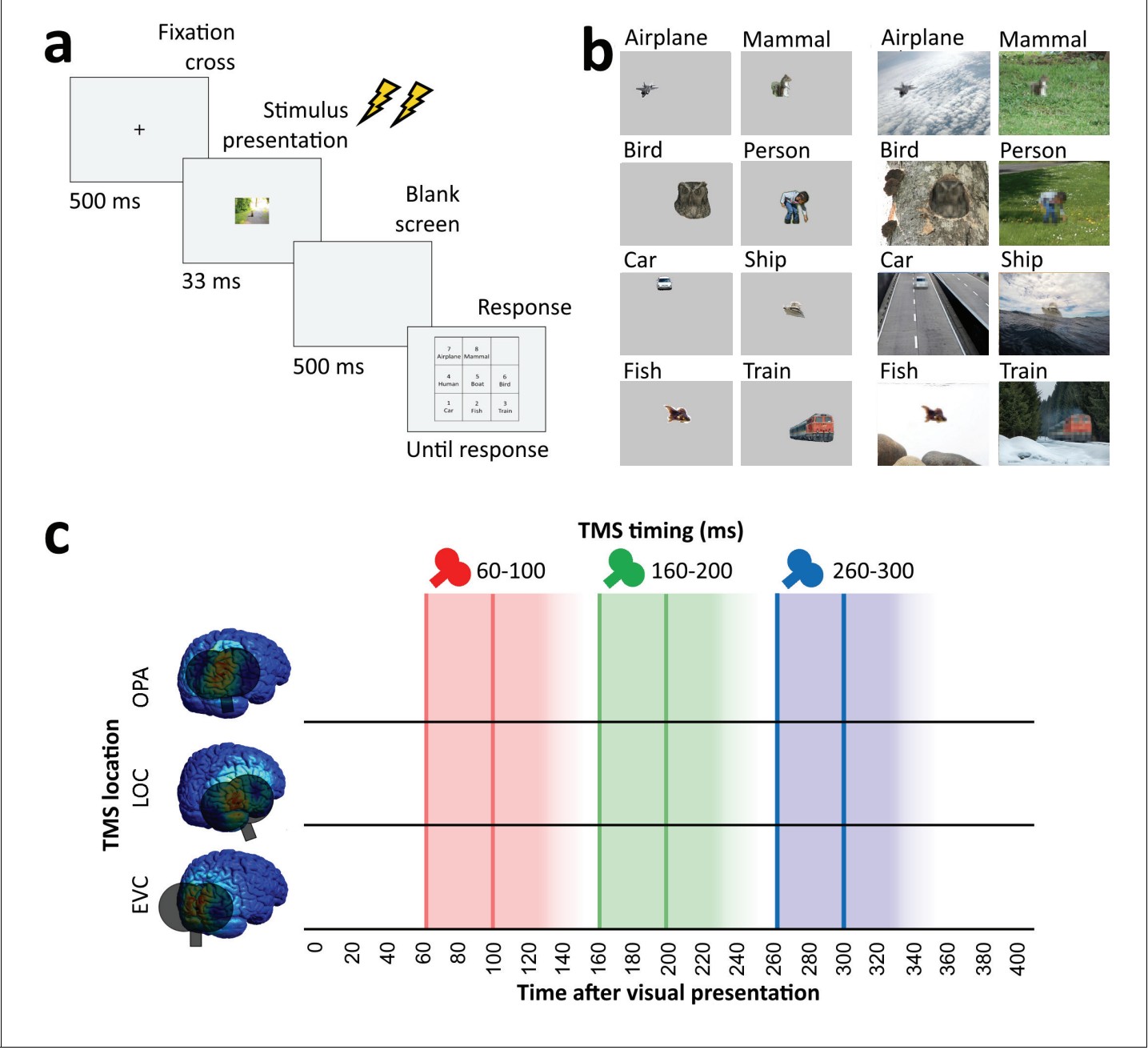

**Figure 2.** Overview of task and stimulation methods. (**a**) Schematic overview of a trial. Two TMS pulses (40 ms apart) were delivered on each trial at one of three time windows relative to stimulus onset (60–100 ms, 160–200 ms, and 260–300 ms). The three TMS timings occurred in random order within each block. (**b**) Examples of each of the eight categories shown in the experiment, in the isolated object condition (left) and the context-based object condition (right). Note that the local degradation of the objects in the context-based object condition is not clearly visible from these small example images. This degradation strongly reduces object recognition when the degraded objects are presented out of scene context (see *Brandman and Peelen, 2017*). These conditions were presented in random order and participants performed the same categorization task on all stimuli. (**c**) Overview of the three TMS sites and the three time windows of stimulation. Shaded background colors indicate presumed time windows of inhibition for double-pulse TMS. TMS, transcranial magnetic stimulation.

performance, recognition in the two object recognition tasks was supported by different neural mechanisms in a time-specific manner (three-way interaction between Task [intact object recognition, context-based object recognition], Region [OPA, LOC, and EVC], and Time [60–100 ms, 160–

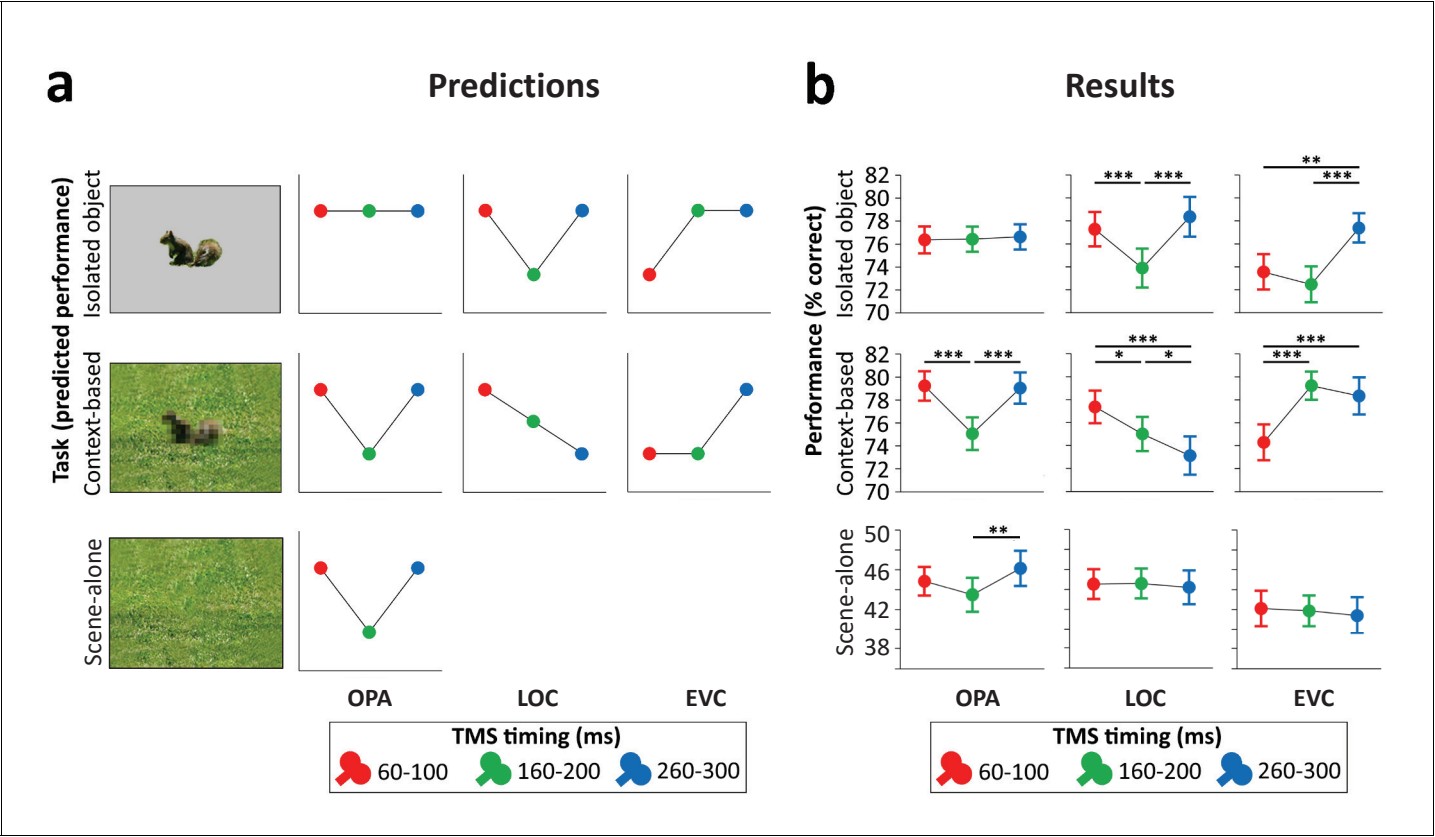

**Figure 3.** Predictions and results. (**a**) We hypothesized that isolated object recognition (top row) would be causally supported by EVC at 60–100 ms (early time point in right plot), followed by LOC at 160–200 ms (middle time point in central plot), reflecting feedforward processing of intact object features (*Cichy et al., 2014*). Scene-selective OPA (left plot) was not expected to contribute to isolated object recognition at any time point (*Dilks et al., 2013*; *Wischnewski and Peelen, 2021*). Similar to isolated object recognition, we hypothesized that context-based object recognition (middle row) would be causally supported by EVC at 60–100 ms and by LOC at 160–200 ms, reflecting feedforward processing. In contrast to isolated object recognition, we hypothesized that OPA would causally support context-based object recognition at 160–200 ms (middle time point in left plot), reflecting scene processing. Crucially, scene-based expectations were hypothesized to reach LOC later in time, disambiguating object representations at 260–300 ms (late time point in central plot; *Brandman and Peelen, 2017*). TMS over LOC at this time point should thus selectively disrupt context-based object recognition. EVC was hypothesized to receive feedback from LOC at 160–200 ms (*Camprodon et al., 2010*; *Koivisto et al., 2011*; *Murray et al., 2002*; *Wokke et al., 2013*), which we expected to be most important for context-based object recognition, in which the object needs to be segregated from the background scene (*Korjoukov et al., 2012*; *Lamme and Roelfsema, 2000*; *Scholte et al., 2008*). Finally, OPA was predicted to causally support scene-alone recognition at 160–200 ms (bottom row). (**b**) Results of three TMS experiments. Predictions were largely confirmed, except for feedback effects in EVC (at 160–200 ms), which were specific to isolated object recognition rather than context-based object recognition. *p<0.05, **p<0.01, ***p<0.001, with error bars reflecting the SEM. EVC, early visual cortex; LOC, lateral occipital cortex; OPA, occipital place area; TMS, transcranial magnetic stimulation.

The online version of this article includes the following source data for figure 3:

**Source data 1.** Individual participant means (accuracy and RT).

200 ms, and 260–300 ms]; F(4,138)=14.37, p<0.001, $\eta_p^2$=0.294). This interaction was followed up by separate analyses for each of the stimulated regions.

TMS did not significantly affect response time (RT), with no interactions involving either TMS time or TMS region: Time×Region, F(4,138)=0.163, p=0.957; Task×Region, F(2,69)=0.81, p=0.450; Time×Task, F(2,142)=0.37, p=0.689; Time×Region×Task, F(4,138)=1.153, p=0.334. There were also no significant main effects of Time (F(2,142)=2.88, p=0.060) or Region (F(2,69)=0.82, p=0.447).

## OPA experiment

Stimulation of scene-selective OPA differentially affected performance in the two tasks (*Figure 3b*, left panel; Task×Time interaction F(2,46)=8.21, p<0.001, $\eta_p^2$=0.263). For *isolated object recognition*, there was no effect of TMS time (F(2,46)=0.07, p=0.935, $\eta_p^2$=0.003), indicating that isolated object

recognition was not influenced by TMS over OPA. By contrast, *context-based object recognition* was strongly modulated by TMS time ($F_{(2,46)}=19.54$, $p<0.001$, $\eta_p^2=0.459$). As predicted, TMS selectively impaired context-based object recognition performance when OPA was stimulated 160–200 ms after scene onset, both relative to earlier stimulation ($t_{(23)}=5.39$, $p<0.001$, $d=1.099$) and relative to later stimulation ($t_{(23)}=5.36$, $p<0.001$, $d=1.095$), with no significant difference between early and late stimulation ($t_{(23)}=0.26$, $p=0.795$). These results show that OPA, a scene-selective region, is causally and selectively involved in (context-based) object recognition.

The pre-registration of the OPA experiment additionally included predictions for a third task, the scene-alone task (*Figure 3a*, bottom row). Similar to the context-based recognition task, we expected that OPA stimulation at 160–200 ms after scene onset would impair accuracy in the scene-alone task. The Task×Time interaction reported above was also significant when including this condition as a third task in the ANOVA ($F_{(4,92)}=4.64$, $p=0.002$, $\eta_p^2=0.168$). For the scene-alone task, accuracy was significantly affected by TMS time ($F_{(2,46)}=4.77$, $p=0.013$, $\eta_p^2=0.172$). TMS impaired scene-alone accuracy when OPA was stimulated at 160–200 ms after scene onset relative to later stimulation ($t_{(23)}=3.02$, $p=0.006$, $d=0.616$), though not relative to earlier stimulation ($t_{(23)}=1.62$, $p=0.118$). There was no significant difference between early and late stimulation ($t_{(23)}=-1.50$, $p=0.145$). Together with the context-based object recognition results, these findings provide information about the causal time course of OPA's involvement in scene recognition, showing a selective OPA effect at 160–200 ms after stimulus onset.

## LOC experiment

Stimulation of object-selective LOC differentially affected performance in the two tasks (*Figure 3b*, middle panel; Task×Time interaction $F_{(2,46)}=12.99$, $p<0.001$, $\eta_p^2=0.361$). For *isolated object recognition*, there was a main effect of TMS time ($F_{(2,46)}=15.50$, $p<0.001$, $\eta_p^2=0.403$; *Figure 3b*). As predicted, TMS selectively impaired isolated object recognition performance when LOC was stimulated at 160–200 ms after stimulus onset, both relative to earlier stimulation ($t_{(23)}=4.58$, $p<0.001$, $d=0.936$) and relative to later stimulation ($t_{(23)}=5.39$, $p<0.001$, $d=1.101$), with no significant difference between early and late stimulation ($t_{(23)}=-1.17$, $p=0.255$). A different temporal profile was observed for *context-based object recognition*. For this task, TMS time also had a significant effect ($F_{(2,46)}=9.03$, $p<0.001$, $\eta_p^2=0.282$; *Figure 3b*). In contrast to the isolated object condition, performance strongly decreased when TMS was applied later in time, at 260–300 ms after stimulus onset, both relative to early stimulation ($t_{(23)}=4.01$, $p<0.001$, $d=0.818$) and relative to middle stimulation ($t_{(23)}=2.26$, $p=0.034$, $d=0.461$). Context-based object recognition accuracy was moderately reduced when TMS was applied at 160–200 ms relative to earlier stimulation ($t_{(23)}=2.17$, $p=0.041$, $d=0.442$). These findings confirm that LOC is causally involved in both isolated object recognition and context-based object recognition at 160–200 ms after stimulus onset. Crucially, LOC was causally involved in context-based object recognition at 260–300 ms, confirming our hypothesis that contextual feedback to LOC supports context-based object recognition.

## EVC experiment

Finally, stimulation of EVC allowed us to test whether similar feedback effects could be observed earlier in the visual hierarchy. Results showed that the time of EVC stimulation differentially affected performance in the two tasks (*Figure 3b*, right panel; Task×Time interaction $F_{(2,46)}=14.42$, $p<0.001$, $\eta_p^2=0.385$). For *isolated object recognition*, there was a main effect of TMS time ($F_{(2,46)}=13.27$, $p<0.001$, $\eta_p^2=0.366$; *Figure 3b*). As predicted, TMS applied early in time impaired recognition performance relative to TMS late in time ($t_{(23)}=3.44$, $p=0.002$, $d=0.701$). Interestingly, and contrary to our prediction, isolated object recognition was also impaired when TMS was applied at 160–200 ms compared to late stimulation ($t_{(23)}=5.19$, $p<0.001$, $d=1.06$). There was no difference in performance between TMS at early and intermediate time windows ($t_{(23)}=1.16$, $p=0.257$). For *context-based object recognition*, there was a main effect of TMS time ($F_{(2,46)}=19.01$, $p<0.001$, $\eta_p^2=0.452$; *Figure 3b*). As predicted, TMS applied early in time impaired recognition performance relative to TMS late in time ($t_{(23)}=5.41$, $p<0.001$, $d=1.105$). Contrary to our prediction, context-based object recognition performance was not significantly reduced when TMS was applied at the middle time window relative to later stimulation ($t_{(23)}=0.99$, $p=0.334$). These findings confirm that EVC is causally involved in initial visual processing, supporting both isolated object recognition and context-based

recognition. In the 160–200 ms time window, EVC was causally involved in isolated object recognition but not context-based object recognition.

## Discussion

Altogether, these results reveal distinct neural mechanisms underlying object recognition based on local features (isolated object recognition) and scene context (context-based object recognition). During feedforward processing, EVC and object-selective cortex supported both the recognition of objects in scenes and in isolation, while scene-selective cortex was uniquely required for context-based object recognition. Results additionally showed that feedback to EVC causally supported isolated object recognition, while feedback to object-selective cortex causally supported context-based object recognition. These results provide evidence for two routes to object recognition, each characterized by feedforward and feedback processing but involving different brain regions at different time points (*Figure 4*).

The finding that EVC (for isolated object recognition) and LOC (for context-based object recognition) causally supported object recognition well beyond the feedforward sweep suggests that feedback processing is required for accurate object recognition. Feedback processing in EVC and LOC may be explained under a common hierarchical perceptual inference framework (*Friston, 2005*; *Haefner et al., 2016*; *Lee and Mumford, 2003*; *Rao and Ballard, 1999*), in which a global representation provides a prior that allows for disambiguating relatively more local information. For context-based object recognition, the scene (represented in OPA) would be the global element, providing a

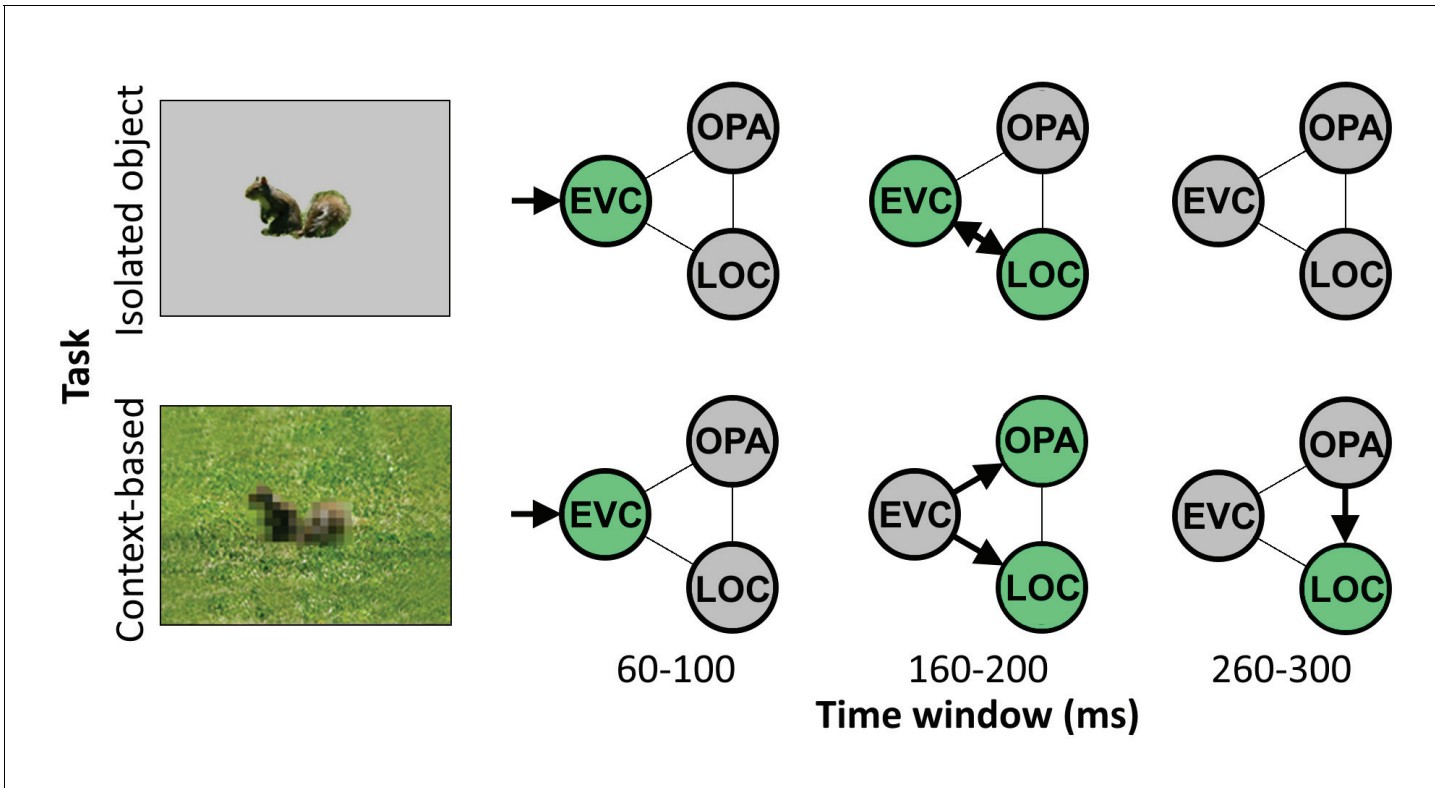

**Figure 4.** Schematic summarizing results. Distinct cortical routes causally support isolated object recognition and context-based object recognition. Isolated object recognition (top row) was supported by EVC early in time (60–100 ms), reflecting initial visual encoding. This was followed by LOC at 160–200 ms, reflecting higher-level object processing. At this time window, EVC was still required for isolated object recognition, presumably reflecting feedback processing. Similar to isolated object recognition, context-based object recognition (bottom row) was supported by EVC at 60–100 ms, followed by LOC at 160–200 ms. However, context-based object recognition additionally required OPA at 160–200 ms, reflecting scene processing. Finally, context-based object recognition causally depended on late processing (260–300 ms) in LOC, reflecting contextual disambiguation (*Brandman and Peelen, 2017*). Note that the arrows do not necessarily reflect direct connections between brain regions. EVC, early visual cortex; LOC, lateral occipital cortex.

prior for processing the relatively more local shape of the object (represented in LOC). For isolated object recognition, object shape would be the global element, providing a prior for processing the relatively more local inner object features (e.g., the eyes of a squirrel; represented in EVC). Feedback based on the more global representations thus serves to disambiguate the representation of more local representations. While feedback processing was hypothesized for LOC based on previous neuroimaging findings, we did not hypothesize that feedback to EVC would be required for recognizing isolated objects. Future studies are needed to test under what conditions feedback to EVC causally contributes to object recognition (*Camprodon et al., 2010*; *Koivisto et al., 2011*; *Wokke et al., 2013*). In line with the reverse hierarchy theory, we expect that the specific feedback that is useful for a given task—and the brain regions involved—depend on the available information in the image together with specific task demands (*Hochstein and Ahissar, 2002*).

An alternative interpretation of the relatively late causal involvement of EVC in isolated object recognition, and LOC in context-based object recognition, is that these effects reflect local recurrence rather than feedback. This interpretation cannot be ruled out based on the current results alone. However, based on previous findings, we think this is unlikely, at least for LOC. In the fMRI study that used a similar stimulus set as used here (*Brandman and Peelen, 2017*), representations of degraded objects in LOC were facilitated (relative to degraded objects alone) by the presence of scene context, indicating input from outside of LOC considering that LOC did not represent object information from scenes presented alone. Furthermore, the corresponding MEG study showed two peaks for degraded objects in scenes, one at 160–180 ms and one at 280–300 ms. The later peak showed a significant contextual facilitation effect in the MEG study, with better decoding of degraded objects in scenes than degraded objects alone. The present finding that TMS over LOC at 260–300 ms selectively impaired context-based object recognition is fully in line with these fMRI and MEG findings, pointing to feedback processing rather than local recurrence.

Taken together with previous findings, the current results are thus best explained by an account in which information from scenes (processed in scene-selective cortex) feeds back to LOC to disambiguate object representations. This mechanism may underlie the behavioral benefits previously observed for object recognition in semantically and syntactically congruent (vs. incongruent) scene context (*Biederman et al., 1982*; *Davenport and Potter, 2004*; *Munneke et al., 2013*; *Võ and Wolfe, 2013*), as predicted by interactive accounts that propose that contextual facilitation is supported by contextual expectations (*Bar, 2004*; *Davenport and Potter, 2004*), with quickly extracted global scene 'gist' priming the representation of candidate objects in the visual cortex (*Bar, 2004*; *Oliva and Torralba, 2007*; *Torralba, 2003*). The current TMS results suggest that OPA is crucial for extracting this global scene information at around 160–200 ms after scene onset, and that this information is integrated with local object information in LOC around 100 ms later. The current results do not speak to whether OPA-LOC connectivity is direct or indirect, for example involving additional brain regions such as other scene-selective regions or the orbitofrontal cortex (*Bar, 2004*).

Our study raises the interesting question of what type of context-based expectations help to disambiguate object representations in LOC. The scenes in the current study provided multiple cues that may help to recognize the degraded objects. For example, the scenes provided information about the approximate real-world size of the objects as well as the objects' likely semantic category. Both of these cues may help to recognize objects (*Biederman et al., 1982*; *Davenport and Potter, 2004*; *Munneke et al., 2013*; *Võ and Wolfe, 2013*). Future experiments could test whether feedback to LOC is specifically related to one of these cues. For example, one could test whether similar effects are found when objects are presented in semantically uninformative scenes, with the scene only providing information about the approximate real-world size of the object.

To conclude, the current study provides causal evidence that context-based expectations facilitate object recognition by disambiguating object representations in the visual cortex. More generally, results reveal that distinct neural mechanisms support object recognition based on local features and global scene context. Future experiments may extend our approach to include other contextual features such as co-occurring objects, temporal context, and input from other modalities.

## Materials and methods

### Participants

Prior to experimentation, we decided to test 24 participants in all three experiments. Preregistrations can be found at https://aspredicted.org/cs4wz.pdf (OPA), https://aspredicted.org/yc969.pdf (LOC), and https://aspredicted.org/cy9fq.pdf (EVC). In total, 72 right-handed volunteers (43 females, mean age ± SD = 23.33 ± 3.59, age range = 18–33) with normal or corrected-to-normal vision took part in the experiment, after participating in a TMS localization experiment (*Wischnewski and Peelen, 2021*). Participants were excluded if they reported to have one of the following: CNS-acting medication, previous neurosurgical treatments, metal implants in the head or neck area, migraine, epilepsy or previous cerebral seizures (also within their family), pacemaker, intracranial metal clips, cochlea implants, or pregnancy. Additionally, participants were asked to refrain from consuming alcohol and recreational drugs 72 hr before the experiment and refrain from consuming coffee 2 hr before the experiment. Participants were divided over three experiments, targeting three cortical areas, based on a previous experiment. All experiments included 24 participants (OPA experiment, 12 females, mean age ± SD = 23.67 ± 3.92; LOC experiment, 14 females, mean age ± SD = 23.50 ± 3.09; EVC experiment, 17 females, mean age ± SD = 22.83 ± 3.81). Prior to the experimental session, participants were informed about the experimental procedures and gave written informed consent. The study procedures were approved by the 'Centrale Commissie voor Mensgebonden Onderzoek (CCMO)' and conducted in accordance with the Declaration of Helsinki.

### Transcranial magnetic stimulation

TMS was applied via a Cool-B65 figure-of-8 coil with an outer diameter of 75 mm, which received input from a Magpro-X-100 magnetic stimulator (MagVenture, Farum, Denmark). Two TMS pulses (biphasic, wavelength: 280 µs) separated by 40 ms (25 Hz) were applied to disrupt visual cortex activity. Given that latency of visual cortex activation varies across participants, a two-pulse TMS design was chosen since it allows for a broader time window of disruption while maintaining relatively good temporal resolution (*O'Shea et al., 2004*; *Pitcher et al., 2007*; *Wokke et al., 2013*). The intensity of stimulation was adjusted to 85% of the individual phosphene threshold (PT). PT was established by increasing stimulator output targeting EVC until 50% of the pulses resulted in the perception of a phosphene while participants fixated on a black screen in a dimly lit room. The TMS coil was placed with the help of an infrared-based neuronavigation system (Localite, Bonn, Germany) using an individually adapted standard brain model over the right LOC, right OPA, or EVC. Each stimulation location was identified through Talairach coordinates set in the Localite neuronavigation system. The coordinates were 45, –74, 0 for LOC (*Pitcher et al., 2009*) and 34, –77, 21 for OPA (*Julian et al., 2016*). TMS was placed on EVC based on its anatomical location, 2 cm above the inion (*Koivisto et al., 2010*; *Pascual-Leone and Walsh, 2001*). We then established the optimal coil position in such a way that phosphenes were reported centrally in the visual field, where the stimuli were presented.

### Experimental stimuli

Stimuli consisted of 128 scene photographs with a single object belonging to one of the following eight categories: airplane, bird, car, fish, human, mammal, ship, and train. For the isolated object recognition task, the object was cropped out of the scene and presented at its original location on a gray background. For the context-based object recognition task, the object was pixelated to remove local features. The experiment additionally included a scene-alone condition, in which the object was cropped out and replaced with background using a content-aware fill tool. In this condition, participants had to guess the object category of the cropped-out object.

To avoid that participants could recognize the degraded objects in scenes based on having seen their intact version, the stimulus set was divided into two halves: for each participant, half of the stimuli were used in the context-based object condition, and the other half of the stimuli were used both in the isolated object condition and the scene-alone condition. This assignment was counterbalanced across participants. The scenes spanned a visual angle of 6°×4.5°.

## Main task

Before the experiment, participants received instructions and were presented with an example stimulus (which was not used in the main experiment). This example displayed how each stimulus variation (context-based, isolated object, and scene alone) was derived from an original photograph. For the main task, each trial started with a fixation cross (500 ms), followed by a stimulus presented for 33 ms. Next, a blank screen was shown for 500 ms. After this, participants were asked to respond by pressing one out of eight possible keys according to the object category presented (*Figure 2*). No limit on RT was given. However, participants were encouraged during the instructions to respond within 3 s. The response screen was presented until the participant responded. The next trial started after a 2 s inter-trial interval. This relatively long interval was chosen to prevent repetitive TMS effects. TMS was applied at one of three different time points, with randomized order. TMS pulses could be applied at 60 ms and 100 ms after stimulus onset, 160 ms and 200 ms after stimulus onset, or 260 ms and 300 ms after stimulus onset. In 2 participants out of the 72 (1 in the LOC experiment and 1 in the EVC experiment), each pulse was accidentally delivered 16 ms earlier than described above.

Each stimulus was repeated three times, once for each TMS timing (60–100 ms, 160–200 ms, and 260–300 ms). This resulted in a total of 576 trials, which were presented in a random order. To avoid fatigue, the task was divided into 12 blocks of 48 trials, each lasting approximately 4 min, with short breaks in between of approximately 1 min. Thus, completing the task took about 60 min. The total duration of the experiment, including preparation and PT determination, was approximately 90 min.

## Acknowledgements

This project has received funding from the European Research Council (ERC) under the European Union's Horizon 2020 Research and Innovation Program (Grant agreement no 725970). The authors would like to thank Talia Brandman for help in stimulus creation, Andrea Ghiani for discussing results, and Marco Gandolfo, Floris de Lange, and Surya Gayet for feedback on an earlier version of the manuscript.

## Additional information

### Competing interests

Marius V Peelen: Reviewing editor, *eLife*. The other author declares that no competing interests exist.

### Funding

| Funder | Grant reference number | Author |
| --- | --- | --- |
| H2020 European Research Council | 725970 | Marius V Peelen |

The funders had no role in study design, data collection and interpretation, or the decision to submit the work for publication.

### Author contributions

Miles Wischnewski, Data curation, Formal analysis, Investigation, Methodology, Project administration, Writing - review and editing; Marius V Peelen, Conceptualization, Supervision, Funding acquisition, Methodology, Writing - original draft, Project administration, Writing - review and editing

### Author ORCIDs

Marius V Peelen https://orcid.org/0000-0002-4026-7303

### Ethics

Human subjects: Prior to the experimental session, participants were informed about the experimental procedures and gave written informed consent. The study procedures were approved by the 'Centrale Commissie voor Mensgebonden Onderzoek (CCMO)' under project number 2019-5311 (NL69407.091.19), and conducted in accordance with the Declaration of Helsinki.

### Decision letter and Author response

Decision letter https://doi.org/10.7554/eLife.69736.sa1
Author response https://doi.org/10.7554/eLife.69736.sa2

## Additional files

### Supplementary files

• Transparent reporting form

### Data availability

All data generated or analysed during this study are included in the manuscript and supporting files. Source data files have been provided for Figure 3.

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
