## [Decision Letter]

**Acceptance summary:**

This study will be of interest to scientists involved in high-level vision. The data provide a compelling demonstration of the causal role of three key visual areas in context-based object recognition. The key claims of the manuscript are supported by the data, and are strengthened by the pre-registration of each of the three experiments.

**Decision letter after peer review:**

Thank you for submitting your article "Causal neural mechanisms of context-based object recognition" for consideration by *eLife*. Your article has been reviewed by 3 peer reviewers, including Redmond G O’Connell as the Reviewing Editor and Reviewer #1, and the evaluation has been overseen by Joshua Gold as the Senior Editor. The following individual involved in review of your submission has agreed to reveal their identity: Peter Kok (Reviewer #3).

Essential revisions:

1) The preregistration for this study commits to including the scene-only condition in the statistical analyses for the main experiment however only the object-based and context-based conditions were considered. The authors should either provide a strong justification for this deviation or else run and report the statistics as originally planned.

2) More detail on the precise participant screening procedure is required. Exactly what tasks and stimuli were participants exposed to? What TMS SOAs were used? How many participants were excluded based on this procedure? A statement should be added to the main text flagging this procedure to the reader so that they are clear that the basic main effect of TMS to LOC object-based task performance was pre-ordained.

3) The authors should provide greater discussion of alternative interpretations of their results. Currently the authors only entertain the possibility that the results reflect feedback effects but they should also consider the possibility of persistent/recurrent activity within visual areas reflecting extended processing of stimuli held in iconic memory.

*Reviewer #1 (Recommendations for the authors):*

The study hypotheses could be articulated more clearly. Citations should be provided to back up the hypothesised TMS SOA effects in each region. The legend of Figure 3 would seem a good place to lay out the predictions in more detail.

The area identified as EVA should be clarified in the Introduction. What visual regions does this area encompass? Just V1? Others?

*Reviewer #2 (Recommendations for the authors):*

1. The authors argue that feedback based on more global representations (of the scene) serve to disambiguate more local representations (of the object).

The scene-only condition seems important to this interpretation, and I would suggest discussing this manipulation further in the main text. An alternative interpretation is that scene recognition simply reduces the epistemic priors of what the object could be (e.g., if the scene is grass, the object is unlikely to be a fish, ship, airplane, or train). Indeed, Figure S2 suggests that performance in the scene-only condition, although relatively poor (40-50%), is much higher than chance (12.5%). So scene recognition seems to filter the range of possible correct responses in the first place, and the effects that are observed may be separate to the capacity of scene recognition to disambiguate specific features of the object itself.

I take the authors' point that the absent effect of TMS on accuracy in the scene-only condition argues against this as a possibility. Nevertheless, an informative condition may have been one in which the image was simply occluded by a shape (e.g., circle) that provided only information about the relative size of the object, but not about its internal features. Such a manipulation would have maintained a similarity with the context-based condition while eliminating the intrinsic perceptual features of the object that could be disambiguated by the scene (other than its approximate size, which provides information about the possible object category, but not the object itself). This may have allowed the authors to more clearly determine whether context-based object recognition is specifically driven by disambiguation of the perceptual features intrinsic to each object.

I am not necessarily suggesting running an additional experiment, but further clarification/discussion on the above issues would be worthwhile.

*Reviewer #3 (Recommendations for the authors):*

– Why where these exact time windows chosen for stimulation (and hypotheses)? Can you provide a more concrete (neurophysiological) rationale?

– The EVC results are opposite to the hypothesised ones, with later involvement for objects in isolation than those in context. The authors acknowledge this, but do not discuss in much depth why this might be. Alternatively, they might simply acknowledge that we do not know why this is, and more research is needed on this point.

– Could you also draw the hypotheses that the parallel account yields, so that it can be clearly seen which data points distinguish the hypotheses? Is the late LOC stimulation effect in context vs. isolation the only crucial point?

– Regarding this datapoint, could it be that late LOC stimulation interferes with object recognition in the context condition because the objects were degraded, rather than because they were presented in a scene context? In other words, because there is more (local) recurrence in LOC required to resolve degraded objects? This seems important to rule out (or acknowledge).

– From Figure S1, it seems that participants were generally faster in the EVC experiment. Any idea why this might be?

---

## [Author Response]

Essential revisions:1) The preregistration for this study commits to including the scene-only condition in the statistical analyses for the main experiment however only the object-based and context-based conditions were considered. The authors should either provide a strong justification for this deviation or else run and report the statistics as originally planned.

The pre-registered statistics for the scene-alone condition in the OPA experiment are now included in the manuscript (p.9-10). The relevant figure (Figure 3) has also been updated such that the scene-alone condition results are now in the main text rather than the Supplement. Results confirmed our predictions, showing a reduction of scene-alone performance when OPA was stimulated 160-200 ms after stimulus onset. Note that the scene-alone condition was only included in the pre-registration of the OPA experiment, which is why we had not reported the corresponding statistics previously. (This condition was not relevant for the LOC and EVC experiments.)

2) More detail on the precise participant screening procedure is required. Exactly what tasks and stimuli were participants exposed to? What TMS SOAs were used? How many participants were excluded based on this procedure? A statement should be added to the main text flagging this procedure to the reader so that they are clear that the basic main effect of TMS to LOC object-based task performance was pre-ordained.

We now introduce the screening procedure in the main text and point the reader to a recent publication that documents the methods and results of this experiment (Wischnewski and Peelen, J Neurosci 2021). The screening experiment followed the design of Dilks et al. (J Neurosci 2013), stimulating OPA and LOC using 5 TMS pulses at a rate of 10Hz (i.e., no SOAs were used). No participants were excluded – all participants were assigned to one of the three conditions (OPA, LOC, EVC). This is now more clearly explained in the manuscript.

3) The authors should provide greater discussion of alternative interpretations of their results. Currently the authors only entertain the possibility that the results reflect feedback effects but they should also consider the possibility of persistent/recurrent activity within visual areas reflecting extended processing of stimuli held in iconic memory.

We have added a paragraph to the Discussion section in which we discuss the alternative interpretation of local recurrence (p.13-14).

Reviewer #1 (Recommendations for the authors):The study hypotheses could be articulated more clearly. Citations should be provided to back up the hypothesised TMS SOA effects in each region. The legend of Figure 3 would seem a good place to lay out the predictions in more detail.

Thanks for these suggestions. We have added citations to back up the hypothesized SOAs. We have also explained the previous fMRI/MEG study that led to these predictions in more detail. We now also included a more detailed explanation in the Figure 3 legend.

The area identified as EVA should be clarified in the Introduction. What visual regions does this area encompass? Just V1? Others?

We have added this clarification to the Introduction, citing previous work using the same procedures. EVC here primarily corresponds to V1.

Reviewer #2 (Recommendations for the authors):1. The authors argue that feedback based on more global representations (of the scene) serve to disambiguate more local representations (of the object).The scene-only condition seems important to this interpretation, and I would suggest discussing this manipulation further in the main text. An alternative interpretation is that scene recognition simply reduces the epistemic priors of what the object could be (e.g., if the scene is grass, the object is unlikely to be a fish, ship, airplane, or train). Indeed, Figure S2 suggests that performance in the scene-only condition, although relatively poor (40-50%), is much higher than chance (12.5%). So scene recognition seems to filter the range of possible correct responses in the first place, and the effects that are observed may be separate to the capacity of scene recognition to disambiguate specific features of the object itself.I take the authors' point that the absent effect of TMS on accuracy in the scene-only condition argues against this as a possibility. Nevertheless, an informative condition may have been one in which the image was simply occluded by a shape (e.g., circle) that provided only information about the relative size of the object, but not about its internal features. Such a manipulation would have maintained a similarity with the context-based condition while eliminating the intrinsic perceptual features of the object that could be disambiguated by the scene (other than its approximate size, which provides information about the possible object category, but not the object itself). This may have allowed the authors to more clearly determine whether context-based object recognition is specifically driven by disambiguation of the perceptual features intrinsic to each object.I am not necessarily suggesting running an additional experiment, but further clarification/discussion on the above issues would be worthwhile.

The scene-alone condition is now included in the main text, as well as in Figure 3. We have also included a more extensive summary of previous fMRI and MEG studies that formed the basis for the current study, where we focused on the question of whether scene and object information are additive or super-additive. We have explained this study in more detail in the Introduction to make the predictions clearer.

The reviewer also raises an interesting point about what aspects the scene helps to disambiguate. Object size is certainly one candidate. We have included a paragraph to the Discussion raising this possibility (p.14-15).

Reviewer #3 (Recommendations for the authors):– Why where these exact time windows chosen for stimulation (and hypotheses)? Can you provide a more concrete (neurophysiological) rationale?

We have added citations to back up the hypothesized SOAs. We have also explained our previous fMRI and MEG work in more detail in the Introduction, as this led to the current predictions.

– The EVC results are opposite to the hypothesised ones, with later involvement for objects in isolation than those in context. The authors acknowledge this, but do not discuss in much depth why this might be. Alternatively, they might simply acknowledge that we do not know why this is, and more research is needed on this point.

We have extended the discussion of these results and mention that more work is needed to follow up on these findings (p.13).

– Could you also draw the hypotheses that the parallel account yields, so that it can be clearly seen which data points distinguish the hypotheses? Is the late LOC stimulation effect in context vs. isolation the only crucial point?

Yes, the late LOC effect is the most informative point to distinguish between the hypotheses. We have now made this clear in the Introduction, also based on a more detailed description of previous fMRI/MEG work.

– Regarding this datapoint, could it be that late LOC stimulation interferes with object recognition in the context condition because the objects were degraded, rather than because they were presented in a scene context? In other words, because there is more (local) recurrence in LOC required to resolve degraded objects? This seems important to rule out (or acknowledge).

We have added a paragraph to the Discussion section in which we discuss the alternative interpretation of local recurrence (p.13-14).

– From Figure S1, it seems that participants were generally faster in the EVC experiment. Any idea why this might be?

There were no significant differences between regions (main effect of Region: (F(2,69) = 0.82, p = 0.447)), which is now reported in the main text (p.8). Thus, while participants were numerically faster in the EVC experiment, this was not reliably different from the other regions (note that Region was manipulated across participants, unlike the other variables).